# *Breathe with the Waves* (BWW)—Creating and Assessing the Potential of a New Stress Management Intervention for Oncology Personnel

**DOI:** 10.3390/curroncol32110632

**Published:** 2025-11-11

**Authors:** Lauren Deckelbaum, Nikita Guarascio, Marie-Pierre Bastien, Anik Cloutier, Maria Kondyli, Marie-Paule Latour, Émélie Rondeau, Serge Sultan

**Affiliations:** 1Department of Psychology, Université de Montréal, Montreal, QC H2V 2S9, Canada; 2Azrieli Research Center, CHU Sainte-Justine, Montreal, QC H3T 1C5, Canada; 3Department of Hematology-Oncology, CHU Sainte-Justine, Montreal, QC H3T 1C5, Canada; 4Department of Pediatrics, Université de Montréal, Montreal, QC H3T 1C5, Canada

**Keywords:** breathwork, oncology, healthcare provider, stress management, intervention

## Abstract

Oncology staff face some of the highest stress levels in healthcare. Stress management programs are often difficult to integrate into daily routines because they require excessive time, are costly, or are not tailored to the realities of oncology personnel. To address these challenges, we developed *Breathe with the Waves* (BWW), a new online program using breathing techniques for stress management. In this study, researchers and health professionals collaborated to create BWW. Twenty oncology professionals completed a questionnaire and were subsequently interviewed. Participants said the program was acceptable, satisfactory and relevant. They reported benefits such as reduced stress, improved work performance, and increased mindfulness. Some challenges, such as discomfort with stress awareness or mild physical reactions, were also noted. Potential outcome measures fell into the following categories: physical and mental health, relational outcomes, work-related outcomes, mindfulness, and personal practice (i.e., how participants may incorporate BWW into their day-to-day activities). These findings suggest that short, accessible programs like BWW could help support oncology personnel.

## 1. Introduction

### 1.1. The Toll of Healthcare Provider Stress

Working in oncology places significant stress on doctors, nurses, allied health professionals, as well as administrative and support staff [1]. Significant sources of occupational stress include an overwhelming workload, poor work–life balance, restricted control over the work environment, excessive administrative requirements, expectations and demands perceived as unrealistic, forced adaptability to ever-changing systems and technologies, emotional overinvolvement with patients and repeated exposure to suffering and death [1,2]. Occupational stress is linked to burnout, characterized by a triad of emotional exhaustion, depersonalization and diminished personal accomplishment [3]. The long-term effects of burnout may lead to severe consequences, including heart disease, anxiety, substance abuse disorders, major depression, stroke and suicide [4]. A study conducted in a Belgian hospital found that oncology staff experienced significantly higher burnout rates than staff in any other department [5]. A survey of over 15,000 physicians from 29 different specialties found that up to 42% of oncology doctors experience burnout [6]. Elevated stress also impacts healthcare systems, leading to retention issues, lower productivity, and reduced efficiency, all of which contribute to increased healthcare costs [7]. A 2019 study estimated that stress-related burnout among U.S. physicians costs the U.S. healthcare system $4.6 billion annually [8]. Moreover, this estimate likely underrepresents the burden, as it excludes downstream effects such as the strain on colleagues covering for absent staff, increased patient dissatisfaction, higher rates of medical errors and greater risk of malpractice lawsuits. Furthermore, numerous studies have demonstrated that patients are at risk of receiving negligent care from stressed providers [9,10,11,12]. Burnout has also been linked to increased patient mortality [13]. Therefore, managing personnel stress should be considered a top priority for oncology leadership.

### 1.2. Stress Management Interventions for Healthcare Professionals

From a public health perspective, stress management should involve large-scale interventions that eliminate the root causes of occupational stress [14]. However, these strategies are difficult to implement in oncology, where many stressors, such as high workloads, exposure to suffering, and emotional involvement with patients, are inherent to the job [15,16]. Large-scale organizational interventions also require a culture shift regarding the expectations placed on doctors, nurses, and support staff, as well as the priority given to the well-being of personnel [15]. As such, most efforts focus on individual-level interventions, though some argue that this approach shifts responsibility from institutions to individuals [17]. While this critique is valid, individual-level interventions currently represent the most feasible near-term option [15]. Accordingly, hospitals tend to offer emotion-focused, rather than problem-focused, coping training [18]. Emotion-focused coping interventions for oncology personnel include cognitive-behavioral therapy, mindfulness-based stress reduction (MBSR), social support groups, yoga, exercise programs, music interventions, general well-being workshops, work–life integration initiatives, and communication skills training [2,19,20,21]. Evidence suggests that these existing programs are associated with promising outcomes [22], yet several drawbacks, such as excessive time demands [20], high costs [23], scheduling conflicts leading to attrition and a shortage of trained instructors [24], limit their uptake and sustainability. As a result, practitioners and researchers have called for stress management programs that are quick, free, accessible, flexible, require minimal training, and can be easily integrated into daily routines [22]. An underused intervention that meets these criteria is controlled breathing.

### 1.3. Breathing

Breathing techniques demonstrate strong therapeutic potential for mitigating the adverse effects of elevated stress [25]. These techniques involve the deliberate regulation of one or more parameters of respiration, such as frequency, depth, or inspiration-to-expiration ratio, to achieve a specific physiological or psychological effect [26]. Numerous studies have shown significant benefits from as little as a few minutes of daily practice [26,27,28,29]. For instance, a recent randomized controlled trial (RCT) with 108 healthy adults found that just five minutes of daily controlled breathing produced greater reductions in stress than five minutes of daily meditation [30]. The immediate physiological effects of breathwork, such as increased vagal tone, may also enhance adherence by making the practice more appealing and easier to sustain compared to meditation. Similarly, a study comparing breathing-focused yoga and meditation-focused yoga found that breathing was more effective for stress reduction and easier for novices to learn and practice [31]. Breathing interventions are simple to master, universally accessible, and can be implemented at little to no cost, require minimal training and can be integrated into daily routines, even for individuals with demanding schedules [25]. Therefore, breathing interventions represent a promising approach for addressing elevated stress in oncology.

### 1.4. Promising Breathing Techniques for Stress Management

Thousands of breathing techniques are used for stress management. A recent attempt to categorize them distinguished four main groups: deep relaxation breathing, mindfulness breathing, holotropic breathing, and yogic breathing [32]. While reviewing every existing method is beyond the scope of this article, we outline three techniques that have gained strong empirical support for stress reduction: cyclic sighing (CS), box breathing (BB), and coherent breathing (CB). Cyclic sighing was included as it may offer the fastest real-time stress relief and can be learned in under a minute [30]. Box breathing was selected for its ease of learning, rapid stress-reducing potential and safety. Finally, coherent breathing was chosen due to its robust evidence base for stress reduction and its ability to increase heart rate variability (HRV), a marker of stress resilience [33]. Notably, no studies have established the minimum duration required for each of these techniques to achieve a stress-reducing effect.

**Cyclic Sighing (CS).** Cyclic sighing consists of two nasal inhalations (one deep, one short/quick) followed by a long exhalation through the mouth, repeated for several rounds [30]. A single cycle is referred to as a “physiologic sigh”. Evidence suggests CS may be the most efficient practice for stress reduction [30]. A single physiologic sigh, performed at any time and under any conditions, can reduce stress and restore calm, making it one of the fastest physiologically verified techniques for acute stress relief [34]. Other research also indicates that when active coping is required, CS outperforms other breathing strategies for stress regulation [35]. To date, however, CS has not been tested among healthcare practitioners, and no studies have specifically examined its use in oncology professionals. This technique is easy to learn, requires minimal training to teach or practice, and can be learned in under a minute [30]. No adverse effects of CS have been demonstrated among healthy adults without psychiatric diagnoses.

**Box Breathing (BB).** Box breathing, also known as “square breathing” or “tactical breathing”, involves equal ratios of inhalation, breath-hold, and exhalation [30]. For instance, one inhales deeply for a set number of seconds, holds the breath for the same duration, exhales for the same duration, and repeats this process. BB is widely used for stress management. A recent study found that five minutes of daily BB over one month produced greater reductions in physiological stress markers than five minutes of meditation [30]. This is a striking finding given the large body of research demonstrating the benefits of meditation for stress. Another study showed that practicing BB for less than 12 min per day over 30 days significantly reduced stress among healthy participants [36]. In performance contexts, novice shooters trained in BB demonstrated lower arousal (attenuated stress) than those without training [37]. Among medical personnel, interventions involving BB (among other stress-reducing techniques) have been shown to be feasible [38]. For instance, BB has been introduced to emergency care providers to mitigate stress and optimize psychological readiness and performance [38,39]. Despite this, BB has not been specifically studied in oncology professionals. It can be practiced independently and requires no extensive training to teach or practice. Some suggest that BB may elicit parasympathetic activation within a single minute [40], though most studies use five-minute sessions [30]. No adverse effects of BB have been reported in healthy individuals [39].

**Coherent Breathing (CB).** Much of the research on breathwork and stress has focused on coherent breathing, also known as resonance breathing [25]. CB involves gentle nose respiration with equal-length inhales and exhales, performed at a rate of about five breaths per minute. This practice optimizes HRV [41,42]; higher HRV is associated with a more resilient stress-response system [43], while lower HRV reflects stress vulnerability [44]. Chronic stress is well documented to reduce HRV [45,46]. CB has been shown to induce calming effects in under five minutes [47]. Interventions using CB have been introduced to reduce stress among military personnel [27], police officers [29], and healthcare providers [33,48]. For example, an RCT found that five weeks of CB training may have prevented nursing students from experiencing increased stress during their first clinical placements, whereas the control group showed heightened stress levels [48]. Another study examined the effects of CB for employees of an academic medical center, including nurses, physicians, patient care assistants and technicians, care coordinators, unit coordinators, administrators, and leadership staff [33]. Reductions in stress and increases in stress resiliency were observed across all roles. To our knowledge, however, no research has evaluated CB specifically among oncology professionals. Importantly, CB requires no extensive training to teach or learn, can be understood in under five minutes and has near-immediate calming effects [43,47]. No significant adverse effects of CB have been reported [41].

Several hypotheses have been proposed to explain the mechanisms underlying the stress-reducing effects of controlled breathing. Physiologically, such techniques may reduce the impact of excess stress by (1) increasing activation of the parasympathetic nervous system [49], (2) decreasing activation of both the sympathetic nervous system and hypothalamic–pituitary–adrenal axis [43], (3) inhibiting overactivation of brain fear centers [50], and (4) strengthening baroreceptor homeostasis [51]. Psychologically, breathing exercises may alleviate stress by changing individuals’ appraisals of stressful events [18], increasing coping resources [52], and fostering a greater perceived sense of control through knowledge and use of coping strategies [53]. Focusing on the breath also parallels mindfulness exercises, shifting focus to the present moment and decreasing attention to stressful thoughts [30].

### 1.5. Purpose of the Present Study

The purpose of this study is to develop and evaluate the potential of a new online stress management program designed specifically for oncology staff. By using evidence-based breathing techniques, this program is designed to be quick, free, accessible and flexible to accommodate busy schedules. It also requires minimal training and targets oncology personnel. To our knowledge, no existing stress management interventions focus specifically on oncology staff while employing breathing exercises alone, without combining them with other activities such as yoga or meditation.

The specific objectives are (1) to design a new online program for oncology personnel that uses evidence-based breathing techniques for stress management, (2) to evaluate the program’s perceived acceptability, satisfaction, and relevance (i.e., social validity) among its target users, (3) to identify perceived benefits and challenges of the program and (4) to generate a list of potential outcomes to be studied in subsequent evaluative trials.

Given their exploratory nature, no hypotheses are proposed for Objectives 1, 3 and 4. For Objective 2, we expect participants to find the intervention acceptable, satisfactory, and relevant. Specifically, scores on the acceptability, satisfaction, and relevance questionnaire are expected to be significantly higher than the neutral midpoint of a seven-point Likert scale. The questionnaire can be found in Appendix A. This study aligns with Phase 1b of the Obesity-Related Behavioral Intervention Trials (ORBIT) model, a systematic framework for developing and evaluating new behavioral interventions [54].

## 2. Materials and Methods

### 2.1. Participants

Objective 1 (creating a new online stress management intervention for oncology personnel) did not require study participants. For Objectives 2, 3 and 4, we recruited a purposive sample of oncology professionals. To ensure adequate representation of the target population (i.e., a diverse group of oncology professionals), participants were drawn from various roles, including doctors, nurses, volunteers, social workers, etc. Consistent with prior studies, individuals were excluded if they self-reported having chronic obstructive pulmonary disease, heart disease, glaucoma, a history of seizures, psychosis, suicidality, bipolar disorder, substance use disorders, or if they were pregnant [30].

Participants were recruited via direct email to hematology-oncology personnel at the CHU Sainte-Justine University Health Center, a large Canadian pediatric hospital, and to volunteers from Leucan, a non-profit association supporting children with cancer and their families. Recruited participants were also asked to invite colleagues who might benefit from the program. The final sample, therefore, consisted of a voluntary, self-selected purposive group. Given the exploratory nature of this study, a sample size of *n* = 20 was deemed sufficient to assess perceived acceptability, satisfaction, and relevance [55], as well as to generate in-depth feedback through semi-structured interviews [56].

### 2.2. Design

We conducted a mixed-methods exploratory study, combining quantitative assessments with qualitative feedback. Acceptability, satisfaction, and relevance were measured through questionnaires, while in-depth feedback was gathered through semi-structured interviews. The study received ethical approval from the CHU Sainte-Justine Research Ethics Committee (#2024-6529).

### 2.3. Materials

Sociodemographic information was collected through a brief questionnaire that covered age, gender, ethnicity, job title, marital status, use of stress-reducing techniques, history of psychological treatment, and current stress levels. The sociodemographic questionnaire can be found in Appendix A.

Acceptability, satisfaction, and relevance were assessed using a 23-item questionnaire adapted to the oncology context. The measure combined the shortened version of the Treatment Evaluation Inventory [57] with a treatment evaluation tool developed by Manne et al. (2016) [58]. Items were rated on a 7-point Likert scale. The validity of this questionnaire has been established in prior research [59]. The questionnaire can be found in Appendix A.

Semi-structured interviews were conducted to further evaluate participants’ feedback on the program, explore perceived benefits and challenges, and generate a list of potential outcomes for future trials (Objective 4). The semi-structured interview guide is available in the Appendix A. Semi-structured interviewing is well-suited to capture perceptions and opinions and enables participants to raise issues meaningful to them [60].

### 2.4. Procedures

#### 2.4.1. Procedure for Objective 1

A core research team was established to design the new program. It included doctoral-level and professional researchers, as well as candidate intervention users (e.g., nurses, doctors, professionals, and volunteers who work in oncology). The precise job titles of candidate users in the research group are listed in Appendix B. Candidate users were included because end-user involvement is often crucial for the successful development of interventions [61]. Researchers in the core design group included a doctoral candidate (LD) and a researcher and professor (SS), who has experience creating interventions for oncology personnel.

**Interventionist.** The intervention instructor (LD) is a doctoral candidate in clinical psychology. She is certified in Breath–Body–Mind™ Fundamentals and has completed the Breath–Body–Mind™ Teacher Training [62]. She has experience teaching breathing techniques for stress reduction to diverse populations, including organizational leaders and refugees.

**The Initial Intervention Script.** Script development followed an iterative process. Candidate users in the research team described typical stressful moments in their workday. These scenarios are presented in Appendix C. We incorporated these scenarios into the initial intervention script to illustrate specific situations where breathing techniques can be useful in oncology. In parallel, we developed instructions for the three chosen breathing techniques (CS, BB, and CB), drawing inspiration from Balban et al. (2023) and the Breath–Body–Mind course material [30,62]. We then translated the script into French to ensure accessibility in both Canadian official languages.

**Transforming the Initial Script into the Final Online Intervention.** The project initiators conducted two meetings with candidate users. In the first meeting, the interventionist delivered the script live and collected open-ended, unstructured feedback to facilitate a bottom-up, exploratory approach [63]. The script was then revised accordingly. Next, the interventionist recorded six videos, one for each breathing technique in English and French, and uploaded them to a secure university website platform. In the second meeting, candidate users reviewed the videos and provided further feedback, which informed final edits. These edited videos became the intervention used for Objectives 2, 3, and 4.

**The Intervention Online.** The six finalized videos were uploaded to Panopto^®^ (Pittsburgh, PA, USA), a common platform used by academic institutions, where videos can be accessed via a secure link. The intervention was hosted online to accommodate the demanding schedules of oncology personnel [22]. Online accessibility enables users to engage with the breathing exercises at any time, and evidence suggests that the time of practice does not impact stress-reducing effects [30]. Practice preferences also vary across individuals [43]. Hosting the program online further promotes accessibility, allowing it to reach a larger number of potential end-users and thereby maximizing its impact [22].

#### 2.4.2. Procedure for Objectives 2, 3 and 4

Participants registered for the study by clicking a link provided in the recruitment email. The link directed them to a digital consent form, a brief sociodemographic questionnaire Appendix A, and a scheduling tool to select a preferred date, time, and location (i.e., in-person at the hospital or via Zoom). For Zoom^®^ (San Jose, CA, USA) sessions, participants were instructed to ensure they were in a quiet, distraction-free space. Once registered, participants received an electronic calendar invite confirming their session details, along with email and text reminders sent one day prior.

Each participant attended a one-hour individual session with the interventionist. Sessions began with a brief welcome and explanation of the agenda. Participants then viewed three instructional videos, starting with CS, followed by BB, then CB, progressing from the shortest to the longest practice to allow them to ease into the exercises. For in-person sessions, videos were projected on a large screen; for remote sessions, participants received video links one at a time via the Zoom^®^ chat. Participants were asked to follow along with the breathing instructions in each video. Following the videos, participants completed a semi-structured interview, recorded using a password-protected application. All raw data were stored securely on a password-protected computer in our research laboratory. After the interview, participants completed the acceptability, satisfaction, and relevance questionnaire, which was administered on the interventionist’s laptop for in-person participants and via a link for remote participants. At the end of the session, participants received a thank-you email containing links to the breathing exercise videos for personal use, along with a $25 Amazon gift card as compensation.

### 2.5. Data Analysis

**Quantitative Data.** Acceptability, satisfaction, and relevance results were analyzed using the SPSS software, version 1.0 (Chicago, IL, USA). Single-sample *t*-tests comparing the mean of each questionnaire item with the theoretical neutral value of 4 on the 7-point Likert scale were conducted. Results were confirmed with non-parametric Wilcoxon rank tests.

**Qualitative Data.** We transcribed the interviews via a secure automatic transcription platform (Sonix, 2025: https://sonix.ai/, San Francisco, CA, USA), followed by a manual review [64]. Participants were identified with numeric codes to ensure anonymity. Transcripts were processed in Microsoft Excel^®^ (Redmond, WA, USA) and analyzed using template analysis [65]. This method was used to provide a flexible yet structured approach to analyzing the data. Two coders (LD, NG) independently coded transcripts from the first 15 participants, applying a priori themes derived from the study aims (i.e., perceived benefits and challenges, as well as outcomes to measure). Once coding no longer produced distinctly new themes or codes, the coders developed an initial template incorporating both the a priori themes and additional codes that had emerged. This template was then applied to the remaining data, with iterative refinements made throughout the process. They took care to ensure that the initial template did not constrain the ongoing analysis. Coders met regularly to discuss the evolving template and to reach a consensus on the inclusion and definition of each code. They also maintained a journal to document changes and justify coding choices. All transcripts were coded, and when new material emerged that was not captured by existing codes, the template was revised, and previously coded transcripts were re-coded accordingly. Given the focus on participants’ subjective experiences with and opinions of the intervention, we used a realist/essentialist paradigm [65]. Only semantic (explicit) themes were explored, rather than latent or interpretive ones. Final themes were identified in response to the research objectives, and a list of potential outcomes to study in future trials was generated.

## 3. Results

### 3.1. Objective 1: Design of BWW

Candidate users in the research team provided contextual examples and rounds of feedback on the intervention iterations, including the initial script and the final recorded videos. The direct output of this step was a new intervention comprising instructional videos teaching three different evidence-based breathing exercises for stress management, specifically designed for oncology personnel. Three videos in English and three in French were created and uploaded to Panopto^® ^(Pittsburgh, PA, USA). The links to all six videos are found in Appendix D. A summary of the intervention, structured according to the Template for Intervention Description and Replication (TIDieR) checklist, is presented in Table 1 [66].

Each BWW video features the same instructor (LD) and begins with an introduction stating the program’s name and its goal of helping oncology personnel manage stress. Exclusion criteria are presented at the start of the first two videos (CS and BB), but are omitted from the third (CB), as that exercise is considered safe for all users to practice. Next, participants are informed that they may watch the videos in any order, though each covers a different exercise. At the outset, participants are told not to teach the exercises to others. They are also reminded that once they have learned the new techniques, they may choose to continue practicing whichever technique(s) best suits them: one, two or all three. Participants are informed that they can practice in any position (sitting, standing, or lying down) and that distraction is normal; they should refocus when they notice their attention drifting. The breathing exercises are then taught. Before and after each breathing practice, participants are instructed to observe their respiration, thoughts, and bodily sensations to notice the impacts of the exercise. Afterwards, practice examples of when and where each exercise might be applied are provided, based on feedback from candidate users and supporting evidence. At the end of each video, viewers are directed to continue with the next video (CS, first, and BB, second) or congratulated for completing the program (CB, third). Time stamps marking the start of each breathing exercise are provided, allowing returning participants to skip introductions.

When training in BWW, participants practice cyclic sighing for one minute, box breathing for three minutes, and coherent breathing for four minutes. Although the minimum duration required for each of these techniques to achieve a stress-reducing effect has not been established, evidence suggests that any practice is better than none [67]. Dose–response analyses on controlled breathing suggest a linear relationship between practice time and stress reduction, with longer practice producing greater benefits. These exercises can be practiced at any time of day [30].

### 3.2. Objective 2: Social Validity of BWW

Overall, participants perceived the new intervention as highly acceptable, satisfactory, and relevant. Verbal responses from the semi-structured interviews supported quantitative results. Acceptability levels were 6.42/7 (SD = 0.34), significantly above the neutral point of 4 (t = 31.871, *p* < 0.001; W = 210.00, *p* < 0.001). Reflecting this, Participant #28a reported seeing “only benefits”. Satisfaction levels were 6.8/7 (SD = 0.3), also significantly above the neutral point (t = 42.891, *p* < 0.001; W = 210.00, *p* < 0.001). Participant #4 described the intervention as “new, pleasant, short and sweet, and exactly what people need: interventions that are quick and simple”. Relevance levels of 6.43/7 (SD = 0.51) were also higher than the neutral point of 4 (t = 21.222, *p* < 0.001; W = 210.00, *p* < 0.001). For example, Participant #20 stated, “I will use the new techniques for my own stress management”.

### 3.3. Objective 3: Anticipated Benefits and Challenges of BWW

Following the principles of template analysis, we developed a final coding template (Appendix A). We synthesized participants’ perceived benefits and anticipated challenges of the new intervention.

The first theme reflected participants’ views on the potential benefits of the program, including stress reduction, improved work performance, and increased mindfulness. The second theme captured the challenges identified by participants, which fell into two categories: experienced and anticipated. Experienced challenges referred to difficulties encountered during the intervention itself, whereas anticipated challenges reflected potential issues participants thought might arise for others.

**Stress Reduction.** Participants emphasized stress management as a key benefit of the program. For instance, when asked what changes might be associated with BWW, Participant #12 stated, “Clearly a reduction in stress”, while Participant #18 described BWW as “a great tool” for stress management at work. Participant #14 believed it could reduce both acute and chronic stress, as well as “improve the quality of life for people who feel really stressed”. Some participants also discussed the potential effects on physiological stress markers, such as heart rate, which a few monitored with smartwatches. As Participant #1 reported, “After each video, I watched it [smartwatch], but really, just out of curiosity, and it was consistent with what you were saying in the videos. Like how it [the exercise taught in the video] can raise the heart rate a bit but then drop it back down [to a lower rate] again [after the exercise]”. Participants suggested that BWW could influence a range of physiological indicators, including heart rate, cortisol levels, blood pressure, and respiration rate.

**Work Performance.** Participants believed the program could improve their efficiency, effectiveness and quality of care. For example, Participant #1, a social worker, explained that practicing the breathing exercises between consecutive meetings could help regain focus, ultimately becoming “a better tool” in their work. Similarly, Participant #13 noted, “Well, of course, if we’re more relaxed, we’re better able to provide care, which means better patient care in the end, and better support for our colleagues”.

**Mindfulness.** Participants also described mindfulness-related benefits. We defined mindfulness as “paying attention in a particular way: on purpose, in the present moment, and nonjudgmentally” [68]. Participant #12 explained that the program fostered a better connection with emotions and noted that taking a break to tune into one’s inner experience could facilitate mindfulness. Similarly, Participant #29 shared, “I feel like I’ve returned, somehow, to the present moment, more easily than before the exercises. It’s like I’m already more grounded in my gestures and my voice. […] It’s like there’s a slowness that has also settled in me, that inhabits me and makes me speak more slowly than usual. It’s… it’s… it’s really a pleasant sensation”.

**Experienced Challenges.** Several participants reported challenges during the program, including dizziness after cyclic sighing, fatigue after completing all three exercises, and discomfort from being aware of their stress levels. For example, Participant #20 shared, “In the first one, if the pace is too fast, it can have an effect. I didn’t get dizzy, but I was close, and my heart was beating faster—even though I was going at my own pace”. Others described feeling tired: Participant #28b explained, “I feel like it kind of drained me”. Moreover, some participants found it difficult to confront their own stress. As Participant #1 reflected, “It’s just that, in the moment, personally, it made me confront the fact that I was really noticing how shallow my breathing was. And then it’s like… ugh… It’s like you’re coming face to face with your stress”.

**Anticipated Challenges. **Participants also identified potential challenges that might arise for others, even if they did not experience them personally. Some suggested that certain personalities may resist mind–body practices similar to breathwork, such as yoga. As Participant #12 explained: “I’ve already seen several people who tell me, ‘Yeah, no, that’s [yoga] really not for me.’ So, for me, it’s not that it would be a potential risk, but I see it more as a potential barrier or limitation—something that might not be applicable to everyone. Uh, yes, some people seem a little more, um, resistant”. Other participants raised safety concerns. For instance, Participant #3 suggested that “for some people who aren’t used to it … they might actually lose consciousness because there’s too much oxygen”. Participant #4 questioned whether participants would be aware enough of their own bodily limitations to follow instructions properly.

### 3.4. Objective 4: Suggested Outcomes for Future Studies

When asked about outcomes to measure in future trials, participants identified several spheres for evaluation (Table 2). The first two were physical and mental health outcomes, including physiological indicators such as heart rate, cortisol levels, blood pressure, or respiratory rate, as well as psychological outcomes like perceived stress and positive emotions. The second sphere was relational outcomes, referring to how the intervention might improve emotional regulation and, in turn, relationships both within and outside the hospital. It includes emotion regulation, whereby participation in the intervention may help individuals better manage their emotional reactions and remain calm in situations that might otherwise provoke distress or overwhelm. The third domain was work outcomes, which refers to how the intervention might improve job performance through greater efficiency, fewer mistakes, and more effective decision-making. The fourth category, mindfulness outcomes, focused on measures of mindful attention and presence, such as the ability to remain in the present moment rather than becoming distracted. The final sphere, personal practice, addressed how participants may integrate the program into their daily lives. It includes tracking when (e.g., mornings, evenings), where (e.g., at home, at work, on public transit), and for how long (e.g., one or five minutes per day) they practice.

### 3.5. Suggestions for Future Use of BWW

Participants also recommended several improvements to BWW. Some suggested adding time stamps in each video to allow users to skip introductions and move directly to the exercises. Others recommended including a clear safety notice at the start, advising against practicing while driving, operating heavy machinery or near a body of water. Opinions differed regarding the amount of spoken instruction: while most participants found it appropriate, some preferred more guidance, while others preferred less, with some even preferring more silence. Generalizable recommendations, rather than individual preferences, were applied to the final version of the intervention.

## 4. Discussion

In this mixed-methods study, we developed *Breathe with the Waves* (BWW), a new online stress management program for oncology staff based on breathing exercises. A purposive sample of 20 oncology professionals rated the program as highly socially valid, i.e., acceptable, satisfactory, and relevant. Participants anticipated benefits in three areas: stress reduction, improved work performance, and mindfulness. They also identified two types of challenges: experienced (e.g., light dizziness after the first exercise, tiredness, and confronting their stress levels) and anticipated (e.g., resistance to mind–body practices and potential safety concerns for some individuals). Finally, participants proposed outcomes for the future evaluation of BWW across six domains: physical health, mental health, relational, work, mindfulness, and personal practice.

### 4.1. Social Validity

Demonstrating social validity, as we did here with BWW, is a crucial first step in evaluating a new intervention [54]. Our results support future refinements and pilot testing. They also align with a recent study demonstrating that a virtual breathing program for healthcare workers was both feasible and well-received [69]. Similarly, other researchers have begun to advocate for the use of breathwork as a stress management tool in this population [70].

### 4.2. Three Categories of Benefits

Three broad categories of benefits emerged: stress reduction, improved work performance, and mindfulness. Regarding stress reduction, participants reported feeling less stressed and more capable of managing future stress after completing BWW, suggesting the program may help them better cope with stressful environments. Although this study did not include pre-post stress measures, some participants wearing bio-trackers (e.g., smartwatches) observed objective changes, such as a reduced heart rate, after the exercises. This observation aligns with existing evidence that breathing exercises can produce measurable physiological changes in stress markers [28,30] and significant changes in perceived stress [25].

The second category of perceived benefits was improved work performance. Participants suggested that BWW could improve their work through three mechanisms: (1) increased self-efficacy, (2) new appraisals of potential stressors, and (3) an increased ability to calm patients. Self-efficacy, defined as the belief in one’s ability to succeed in specific situations, is a predictor of job performance [71]. Evidence from an RCT of a web-based stress management intervention for highly stressed employees indicated that increased self-efficacy mediated the program’s beneficial effects on stress reduction [72]. A second mechanism identified by participants involved changing their appraisals of potential stressors. Participants suggested that breathing exercises might alter how individuals perceive and interpret challenges, a notion that has been supported [73]. Research has also shown that having greater coping resources can positively influence stress appraisals and, in turn, work performance [74]. A third mechanism participants suggested was that BWW might indirectly improve patient care by helping staff remain calm in stressful interactions. This idea is supported by evidence indicating that calm provider behavior is linked to improved patient well-being [75], reduced provider stress is associated with higher patient satisfaction [76], and provider stress levels correlated with patient distress [77]. Provider stress can manifest as irritability or detachment, and patients who perceive their providers as emotionally distant may experience greater distress [78]. If future research confirms the effects of BWW, these pathways could be explored.

The final category of benefits was increased mindfulness. Existing research suggests that controlled breathing may impact mindfulness [25], particularly through its effects on attention [22]. Breathing exercises provide a clear object of focus, anchoring attention to the present moment and reducing mind-wandering or rumination, core aspects of mindfulness [68]. Biological evidence also supports the role of breathwork in improving present-moment attention and decreasing mind-wandering [22]. Thus, participants’ reports of greater mindful awareness following BWW are consistent with existing literature.

### 4.3. Two Types of Challenges

Participants identified experienced and anticipated challenges. Regarding experienced challenges, some participants reported mild dizziness after the cyclic sighing exercise. Although uncomfortable, this reaction is not uncommon or unsafe, provided the exclusion criteria are respected and participants avoid exceeding their personal limits [30]. BWW included instructions on what to do if dizziness occurred. Another reported challenge was tiredness after the third video, which taught coherent breathing. While this could be seen as a risk in some contexts, it may also represent a benefit in others. The evidence is mixed: an RCT with 400 participants found that CB did not induce sleep [25], whereas other studies suggest that it promotes relaxation [26]. Interestingly, this relaxation effect may not exceed that of placebo-paced breathing, suggesting that any controlled breathing may help relax [25]. The fatigue reported here may reflect the cumulative impact of practicing all three breathing techniques in succession or an increased self-awareness of pre-existing tiredness. Another experienced challenge was facing one’s true stress levels. One participant explained that the intervention made them realize how stressed they had been before the intervention, a realization they found unsettling. This highlights a potential rebound effect, where participants who were previously unaware of their inner state become uncomfortably conscious of it after the intervention. However, research suggests that avoidance of difficult states such as stress, including denial, can be harmful in the long term, despite being more comfortable in the short term. For example, a 10-year longitudinal study of over 1200 adults found that reliance on avoidance strategies was associated with higher stress over time [79].

Besides dizziness, fatigue, and heightened awareness of stress levels, participants also anticipated other challenges. First, some questioned whether individuals with negative attitudes toward mind–body practices such as breathwork would engage with the program. Research supports this concern, such that personality traits influence both participation in and attitudes toward mind–body interventions. For example, people high in openness to experience are more likely to adopt mindfulness practices, while those lower in openness may be less inclined [80]. Second, participants noted potential safety concerns. BWW instructions emphasize respecting personal limits (e.g., “pay attention to how your body feels and stop if the exercise feels too uncomfortable for you”), a common precaution in mind–body programs [30,68]. However, as one participant observed, this assumes a baseline ability to recognize and respond to bodily cues. Research shows that some populations, such as trauma survivors or individuals with eating disorders, may dissociate from these cues and therefore struggle to follow such instructions [81]. Further research is needed to understand this possibility.

### 4.4. Future Directions

The present exploratory study created a new program, *Breathe with the Waves *(BWW), which was found to be acceptable, satisfactory, and relevant among 20 oncology personnel. Participants identified the program’s benefits and challenges, and a detailed list of potential outcomes for future research was generated. The next step is to establish the feasibility of BWW with the target population, followed by preliminary proof-of-concept testing. For example, future studies could measure changes in perceived stress before and after participation. If promising, more rigorous pilot trials using randomized designs and larger, more diverse samples should be undertaken.

Notably, our findings suggest that breathing exercises do not require long practice durations to be perceived as beneficial. The active practice periods in BWW were shorter than those used in many previous studies, highlighting the need to determine the minimum duration required for breathing exercises to achieve a stress-reducing effect. Another avenue for future research is to examine the role of breathwork within broader mind–body interventions.

Moreover, the intervention’s elements, namely its brevity, accessibility and flexibility, make it well-suited for oncology personnel. Designed to be quick, free and easy to integrate into demanding schedules with minimal training, BWW could also benefit healthcare professionals in other high-stress environments. Future research should therefore explore adapting and evaluating BWW for use across diverse high-stress healthcare settings.

### 4.5. Strengths and Limitations

A key strength of this study is the iterative, co-constructive development of BWW. Multiple rounds of feedback from candidate end-users and researchers ensured the intervention was adapted to the real-world contexts of oncology practice. This process reflects principles of co-construction, where end-users collaborate to shape interventions by contributing to content, format and implementation strategies. Evidence from healthcare research suggests that co-constructing interventions with end-users can improve their acceptability and contextual fit, thereby making interventions more feasible and relevant to the individuals they aim to support [61,82]. In this study, we believe engaging a diverse group of oncology professionals in the development process, including nurses, doctors, professionals, and volunteers, helped ensure the program addressed the specific needs of its target users.

We should acknowledge several limitations. First, participants were self-selected and may already have had an interest in mind–body interventions. If the sample was randomly selected, we might have observed lower social validity ratings. Second, because the interventionist also conducted the semi-structured interviews, participants may have been reluctant to share negative feedback, potentially inflating positive evaluations. Future studies should use independent evaluators to reduce this bias.

## 5. Conclusions

In this mixed-methods exploratory study, we developed *Breathe with the Waves *(BWW), a new online stress management program for oncology staff. The intervention is publicly available for future refinement and research, with both a TIDieR checklist and links to the videos provided. A purposive sample of 20 oncology professionals rated BWW as highly acceptable, satisfactory, and relevant. Participants identified three categories of potential benefits (i.e., stress reduction, improved work performance and mindfulness), and two categories of challenges: experienced (e.g., dizziness, fatigue, confronting stress levels) and anticipated (e.g., safety risks or resistance from certain personality types). They also generated a list of potential outcomes for future studies, including physical and mental health, relational aspects, work-related factors, mindfulness, and personal practice. Feasible, accessible stress management interventions for oncology personnel remain scarce. BWW represents an important first step toward addressing this critical need.

## Figures and Tables

**Table 1 curroncol-32-00632-t001:** Definition of *Breathe with the Waves*: a new stress management intervention for oncology personnel, according to the main criteria of the TIDieR checklist [66].

Criterion	Explanation
Name of the intervention	*Breathe with the Waves* (BWW)
Why (Rationale)	Breathing exercises, particularly (1) cyclic sighing, (2) box breathing, and (3) coherent breathing, are evidence-based techniques shown to reduce stress. Breathing interventions have also been found feasible in healthcare settings.
What (Materials)	Six pre-recorded instructional videos (three in English and three in French) accessible via internet link. The videos teach the three breathing exercises and are tailored for oncology personnel. - Cyclic Sighing (https://umontreal.ca.panopto.com/Panopto/Pages/Viewer.aspx?id=a182c984-fa69-42e0-aca1-b33701730237) (accessed on 19 August 2025) (English version) - Cyclic Sighing (https://umontreal.ca.panopto.com/Panopto/Pages/Viewer.aspx?id=fdb790ea-d317-418f-81fe-b337013d3181) (accessed on 19 August 2025) (French version) - Box Breathing (https://umontreal.ca.panopto.com/Panopto/Pages/Viewer.aspx?id=fd014d43-c6a5-486a-9c28-b337017dbfda) (accessed on 19 August 2025) (English version) - Box Breathing (https://umontreal.ca.panopto.com/Panopto/Pages/Viewer.aspx?id=2a4b0615-a98b-471a-86e9-b33a00076f29) (accessed on 19 August 2025) (French version) - Coherent Breathing (https://umontreal.ca.panopto.com/Panopto/Pages/Viewer.aspx?id=086f2c6b-d2e0-4556-ab5b-b338000c7ce3) (accessed on 19 August 2025) (English version) - Coherent Breathing (https://umontreal.ca.panopto.com/Panopto/Pages/Viewer.aspx?id=21f31ed0-bac9-4f7d-97b0-b337013d5027) (accessed on 19 August 2025) (French version)
What (Procedures)	Participants watch each video, following the instructions as they are presented. Videos can be viewed in any appropriate setting (e.g., at home or at the hospital).
Who Provided (Interventionist)	The intervention is self-guided; participants watch the videos and practice independently. No live interventionist is needed.
How (Mode of delivery)	The intervention is accessible online via the Panopto^®^ platform (Pittsburgh, PA, USA) of the Université de Montréal. The main activity is to watch three videos and follow the instructions in real time.
Where	The intervention can be practiced in any safe setting, except while driving, operating heavy machinery, or near a body of water.
When and How Much	We recommend that participants first use BWW to learn each breathing exercise (CS, BB, CB), then practice them as needed, with or without the videos. Participants can watch the videos in any order, though the recommended sequence is Video 1, followed by Videos 2 and 3.
Tailoring	BWW is a flexible tool that can be used intensively or when needed. Once participants learn the new techniques through BWW, they may use all three techniques or select the one(s) that best suit their preferences.

**Table 2 curroncol-32-00632-t002:** Outcomes suggested by participants to be explored in future research on BWW.

Category	Outcome to Measure	Example Citation from Transcript
Physical health outcomes	Heart rate	Participant #5: *Heart rate.*
	Cortisol level	Participant #11: *Biological things that could be measured, like cortisol.*
	Blood pressure	Participant #11: *Blood pressure.*
	Respiratory rate	Participant #13: *Maybe respiratory rate.*
Mental health outcomes	Positive emotions	Participant #29: *I feel lighter, I can feel it. It’s like there’s a feeling… like a wave passed through and took away the things that were bothering my mind. And it’s like I came back to the present moment more easily than before the exercises. I feel like I’m already more grounded in my movements and my voice. It’s as if a sense of calm has settled in. So, it’s a feeling of well-being and comfort—cozy, gentle, restful. It’s like a slowness has taken hold of me, inhabits me, and makes me speak more slowly than usual. It’s… it’s… it’s really a pleasant sensation.*
	Perceived stress reduction	Participant #12: *Yeah. Clearly, a reduction in stress* [is caused by the intervention].
		Participant #14: […] *If we manage to do these exercises regularly enough, I think it can improve the quality of life for people who feel really stressed.*
	Feeling relaxed	Participant #20: *It really works. It relaxes you. It’s incredible. I feel so relaxed right now.*
	Feeling energized	Participant #13: *When I said my breathing was less shallow, it allowed me to *[…] *lift my head above water and say, “Alright, let’s go. Let’s do this!”*
Relational outcomes	Emotion regulation	Participant #1: *Emotion regulation or management.*
Work outcomes	Improved performance	Participant #1: […] *and even better efficiency at work. I think it can, you know, when you just can’t seem to write your report, or when you know you’ve had back-to-back meetings with… You know, in my case, as a social worker, I meet with parents. I can have several meetings in one day. Between two meetings, becoming… how can I say… a better version of myself, we always say that we’re our own tool, so becoming a better tool.*
Mindfulness outcomes	Mindfulness	Participant #12: […] *And also a better connection with feelings… On the mindful aspect, I would say taking a pause and trying to observe, to listen a bit more to your own inner voices. It’s something that can be facilitated by this kind of exercise.*
Personal practice	Tracking	Participant #28a: *Measure how we will apply it.*
		Participant #28b: *Post, maybe we should practice it and then give more feedback after we’ve practiced it for a while to see if we manage to implement it.*

## Data Availability

The data presented in this study are available on request from the corresponding author. The data are not publicly available due to privacy and ethical restrictions.

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
