# Peer review of "Breathe with the Waves (BWW)—Creating and Assessing the Potential of a New Stress Management Intervention for Oncology Personnel"

_curroncol, 2025, doi:10.3390/curroncol32110632_

Round 1

Reviewer 1 Report

Comments and Suggestions for Authors

Thank you for the opportunity to review this well written manuscript. I found it timely and helpful, I suspect reading the report takes almost less time than the intervention.

I found two minor errors and have one suggestion. On line 283 the word "affect" is used. I'm not sure you mean that in a unidirectional manner e.g., lessen or decrease, or that there was no change (which could be bidirectional)? Reference #57 has errors in the formatting.

Finally the question. You present a good case for creating an intervention specifically for oncology staff. However, the elements of the intervention could be used by other health care professionals working in a high stress environment. I would suggest noting that possibly as a serendipitous finding in your discussion. 

Serendipitous

Author Response

Thank you for reviewing our manuscript. We are pleased to hear your largely positive feedback supporting our study's contribution to the empirical literature. We have considered each of your suggestions and comments, and we have modified our manuscript accordingly. Please find the detailed responses below and the corresponding revisions/corrections highlighted in yellow in the re-submitted files.

Comment 1: On line 283, the word “affect” is used. I’m not sure you mean that in a unidirectional manner, e.g., lessen or decrease, or that there was no change (which could be bidirectional)?

Response 1: Thank you for pointing this out. This was a typographical error. We replaced the word “affect” with “impact.” The correction was made on lines 284-285 of page 7.

Comment 2: Reference #57 has errors in the formatting.

Response 2: Thank you for bringing this to our attention. We corrected the formatting errors in Reference #57 by aligning the reference. This modification was made on lines 833-834 of page 21 of the manuscript.

Comment 3: You present a good case for creating an intervention specifically for oncology staff. However, the elements of the intervention could be used by other health care professionals working in a high stress environment. I would suggest noting that possibly as a serendipitous finding in your discussion.

Response 3: Thank you for your comment. We added your suggestion as a potential avenue for future research, noting that BWW could also benefit healthcare professionals in other high-stress environments. This modification was made on lines 598-603 of page 16 of the manuscript.

Reviewer 2 Report

Comments and Suggestions for Authors

Title: Breathe with the Waves” (BWW) – Creating and Assessing the Potential of a New Stress Management Intervention for Oncology Personnel General Assessment

This manuscript presents the development and preliminary evaluation of Breathe with the Waves (BWW), an online breathing-based stress-management program designed for oncology professionals. The topic is highly relevant, given the persistent problem of burnout and stress in oncology care. The authors effectively explain the need for a time-efficient and low-cost intervention for oncology staff.

The manuscript is well organized and concise. Developing a bilingual (English and French) digital tool is another strength, increasing accessibility. The authors did well to report both benefits (reduced stress, improved mindfulness) and challenges (discomfort, mild physical reactions), which adds transparency and realism to the findings.

The introduction is comprehensive, logically structured, and highly informative. It successfully establishes the background, defines the problem of stress among oncology personnel, reviews existing intervention gaps, and builds a strong rationale for developing the Breathe with the Waves (BWW) program.

The methodology is comprehensive and well-articulated, showing careful planning.

The use of clear subheadings (Objectives 1–4) enhances readability in Results section. Although rich in detail, some subsections—particularly Objective 3—are quite lengthy. Consider condensing repetitive quotes or summarizing certain participant remarks in paraphrased form to improve readability and flow. Table 1 and Table 2 are informative but visually dense. Consider arranging them more reader-friendly.

Overall, I find this to be a promising and well-articulated manuscript.

Author Response

Thank you for reviewing our manuscript. We are pleased to hear your largely positive feedback supporting our study's contribution to the empirical literature. We have considered each of your suggestions and comments, and we have modified our manuscript accordingly. Please find the detailed responses below and the corresponding revisions/corrections highlighted in yellow in the re-submitted files.

Comment 1: Although rich in detail, some subsections–particularly Objective 3–are quite lengthy. Consider condensing repetitive quotes or summarizing certain participant remarks in paraphrased form to improve readability and flow.

Response 1: Thank you for your suggestion. We condensed the written results for Objective 3 by removing repetitive quotations and paraphrasing several participant remarks. This revision was made on lines 395-462 of pages 10-11 of the manuscript.

Comment 2: Table 1 and Table 2 are informative but visually dense. Consider arranging them more reader-friendly.

Response 2: Thank you for your comment. For Table 1, we recognize the need for more reader-friendly material; however, the TIDieR criteria are an essential aspect of intervention description and, in essence, are descriptive. Regarding Table 2, we explored alternative formats; however, these resulted in a presentation that was less systematic and less clear.

Round 2

Reviewer 2 Report

Comments and Suggestions for Authors

Thanks for your replies.